# Hemoglobin β Expression Is Associated with Poor Prognosis in Clear Cell Renal Cell Carcinoma

**DOI:** 10.3390/biomedicines11051330

**Published:** 2023-04-30

**Authors:** Yuta Kurota, Yuji Takeda, Osamu Ichiyanagi, Shinichi Saitoh, Hiromi Ito, Sei Naito, Hironobu Asao, Norihiko Tsuchiya

**Affiliations:** 1Department of Urology, Faculty of Medicine, Yamagata University, 2-2-2, Iida-Nishi, Yamagata 990-9585, Japan; 2Department of Immunology, Faculty of Medicine, Yamagata University, 2-2-2, Iida-Nishi, Yamagata 990-9585, Japan

**Keywords:** hemoglobin β-chain (HBB), malignant transformation, redox balance, renal cell carcinoma (RCC)

## Abstract

Background: The regulation of the redox balance in the tumor microenvironment is thought to be an adaptive response of tumor cells to hypoxic environments. In recent years, it has been reported that the hemoglobin β-chain (HBB), which is involved in scavenging reactive oxygen species (ROS), is expressed in several carcinomas. However, the relationship between HBB expression and the prognosis of renal cell carcinoma (RCC) remains unclear. Methods: HBB expression was immunohistochemically analyzed in 203 nonmetastatic clear cell RCC (ccRCC) cases. Cell proliferation, invasion, and ROS production were measured in ccRCC cell lines treated with HBB-specific siRNA. Results: The prognosis of HBB-positive patients was worse than that of HBB-negative patients. Cell proliferation and invasion were inhibited, and ROS production was increased by treatment with HBB-specific siRNA. Oxidative stress increased HBB expression in cells exposed to H_2_O_2_. Conclusions: HBB expression in ccRCC contributes to cancer cell proliferation by suppressing ROS production under hypoxic conditions. Taken together with clinical results and in vitro experiments, HBB expression may serve as a new prognostic biomarker for RCC in the future.

## 1. Introduction

The identification of malignant tumor factors is important for enhancing therapeutic responsiveness and discovering new therapeutic targets. Recently, it has been demonstrated that the malignant transformation of cancer is triggered by hypoxic conditions in the tumor microenvironment [1]. Reactive oxygen species (ROS) production increases under hypoxic conditions owing to a decline in mitochondrial function, and the malignant transformation of tumor cells requires adaptation to ROS [2,3]. Increased expression levels of various scavengers as an adaptive strategy against increasing ROS levels have been reported. Among these molecules, hemoglobin β (HBB) is a protein with ROS-scavenging functions [4,5].

The red blood cells of vertebrates, including humans, and some invertebrates contain the protein hemoglobin. The main function of hemoglobin is to transport oxygen from the lungs or gills to various tissues. Normal hemoglobin is a tetramer comprising two α-globin chains and two β-globin chains (hemoglobin β, HBB), and each chain contains a heme-binding site for oxygen transport. Recently, HBB was found to be expressed in tissues other than erythrocytes, such as alveolar epithelial cells [6,7,8], retinal pigment epithelial cells [9], mesangial cells [10], hepatocytes [11], nerve cells [12,13,14], and cervical epithelial cells [4]. These reports suggest that the physiological functions of HBB include promoting intracellular oxygen uptake and protecting cells from oxidative damage.

HBB is also expressed in cancer cells, such as breast and lung cancer cells [15,16]. However, studies on the relationship between HBB expression and prognosis have been limited to DNA microarray data analysis from published databases [15,16]. Furthermore, the involvement of HBB expression in the malignant transformation of cancer remains unclear.

The most common malignant kidney tumor is renal cell carcinoma (RCC), and the clear cell (cc) type accounts for 80% of histological types [17,18]. RCC incidence and mortality rates are rising worldwide, including in Japan [17,18]. Approximately 20–30% of patients with RCC present with metastatic disease at the time of initial diagnosis, and up to 30% of the patients who undergo curative surgery for clinical-confined RCC suffer from disease recurrence [17,18,19]. Although the survival rate of patients with metastatic RCC has greatly improved with the advent of molecularly targeted drugs and immune checkpoint inhibitors [20], the 5-year survival rate of patients with advanced RCC (T3-4, N+, M+) is less than half [17]. As with other carcinomas, the discovery of prognostic indicators and the elucidation of molecular mechanisms that can serve as new therapeutic targets are critical.

In this study, we investigated the relevance of HBB expression in RCC clinical outcomes. Furthermore, we examined the function of HBB in RCC cell lines to validate the potential of HBB expression as a prognostic biomarker.

## 2. Materials and Methods

### 2.1. Patients

This study was approved by the Yamagata University Faculty of Medicine Ethics Committee (No. H29-15). Informed consent was obtained from each patient, in compliance with the ethical standards of the Declaration of Helsinki. We retrospectively explored the medical archives of consecutive patients who underwent curative surgery for non-metastatic (M0) ccRCC at Yamagata University Hospital between 2000 and 2010. The inclusion criteria were as follows: (1) pathologic diagnosis of ccRCC using resected specimens obtained via radical or partial nephrectomy, and (2) availability of formalin-fixed paraffin-embedded specimens. The exclusion criteria were as follows: (1) genetic disorders that increase the risk of ccRCC, such as von Hippel–Lindau disease; (2) synchronous bilateral ccRCC; and (3) administration of molecular-targeted drugs for neoadjuvant and/or adjuvant therapy. Two hundred and three Japanese patients with non-metastatic ccRCC were eligible, and clinical and pathological data were collected from the patients’ medical records. The tumors were fixed in 10% buffered formalin, embedded in paraffin, and the samples were coded. Paraffin sections were routinely stained with hematoxylin and eosin, and a pathological diagnosis was made as previously described [21]. Pathological evaluation was performed according to the 2016 World Health Organization classification [22] and the 2017 American Joint Committee on Cancer Tumor-Node Metastasis staging system. 

### 2.2. Cell Lines

Four ccRCC cell lines (A498, Caki1, 769-P, and 786-O) were obtained from the American Type Culture Collection (Manassas, VA, USA). The cells were cultured in RPMI1640 medium (Thermo Fisher Scientific, Waltham, MA, USA) containing 50 μg/mL kanamycin and 10% fetal bovine serum (FBS) under 5% CO_2_ at 37 °C in high humidity conditions using a CO_2_ incubator.

### 2.3. Immunohistochemical Observation

The formalin-fixed and paraffin-embedded tissue was sliced into 3 μm thick slices. After removing the paraffin by xylene immersion, samples were hydrated from 100% ethanol to pure water. Antigen retrieval was performed by soaking in 10 mM citrate buffer (pH 6.0) and autoclaving at 120 °C for 20 min. Inhibition of endogenous peroxidase was performed by immersion in methanol containing 3% hydrogen peroxide for 15 min. Blocking for non-specific antibody reactions was performed with phosphate-buffered saline (PBS; 8 mM Na_2_HPO_4_, 1.4 mM KH_2_PO_4_, 136 mM NaCl, 5 mM KCl, pH 7.4) containing 1% bovine serum albumin for 30 min. To detect HBB expression, horseradish peroxidase-labeled mouse anti-human HBB antibody (1:1200, sc-21757 HRP, Santa Cruz Biotechnology, Dallas, TX, USA) was added at 20–25 °C for 60 min. After washing with PBS, Histofine Simple Stain (MAX-PO[M]; Nichirei, Tokyo, Japan) was used to amplify the antibody reaction (20–25 °C for 30 min). To develop color, diaminobenzidine (D5905, Sigma-Aldrich, St. Louis, MO, USA) was added for 1 min. Nuclear staining was performed using hematoxylin. Erythrocytes from the normal renal glomeruli were used as positive controls. 

When blood vessel-derived red blood cell invasion into tumor tissue was excluded from the HBB-positive area and the area of HBB expression (brown-colored area) in tumor tissues was 5% or more, it was defined as an HBB-positive case. The diagnosis of HBB-positive cases was made by YK, who was blinded to the patient data.

### 2.4. Measurement of HBB Expression Using Flow Cytometry 

Cultured cells were detached from plastic dishes using a Trypsin-EDTA solution (T4049, Sigma-Aldrich) and fixed with PBS containing 1% formalin for 15 min at 20–25 °C. After fixation, the cell membrane was permeabilized using PBS containing 0.1% TritonX-100 for 5 min at 4 °C. After washing with 1% FCS-containing PBS, FITC-conjugated mouse anti-human HBB antibodies (sc-21757 FITC, Santa Cruz Biotechnology) were applied for 30 min at 4 °C. The cells were washed with PBS followed by suspension in PBS containing 1% formalin until measurement with a flow cytometer (FACSCanto II; BD Biosciences, Franklin Lakes, NJ, USA). Data were analyzed using FlowJo software version 6.5.7 (TreeStar, Ashland, OR, USA). For negative control staining, FITC-conjugated mouse IgG1 antibody (MOPC-21, BD Biosciences) was used as the isotype control antibody.

### 2.5. HBB mRNA Measurement by Quantitative Real-Time Reverse Transcript (RT) Polymerase Chain Reaction (PCR)

RNA was purified using TRIzol Reagent (Invitrogen, Carlsbad, CA, USA), and cDNA was synthesized using the ReverTra Ace qPCR RT Master Mix with a gDNA Remover kit (Toyobo, Osaka, Japan). cDNA was amplified by PCR using the TB Green Fast qPCR Mix (Takara Bio Inc., Tokyo, Japan) and quantified using a CFX96 Touch real-time PCR analysis system (BioRad Laboratories, Hercules, CA, USA). PCR conditions were as follows: initial denaturation at 95 °C for 2 min, denaturation at 95 °C for 10 s, and 45 cycles of annealing/extension at 58 °C for 30 s. The expression levels were quantified using the comparative Ct method as previously described [23]. Glyceraldehyde-3-phosphate dehydrogenase (GAPDH) was used as an endogenous control. The HBB primer pair sequences were 5′-GTG CTC GGT GCC TTT AGT GA-3′ and 5′-AGC GAG CTT AGT GAT ACT TGT GG-3′. The GAPDH primer pair sequences were 5′-GCA CCG TCA AGG CTG AGA AC-3′ and 5′-TGG TGA AGA CGC CAG TGG A-3′.

### 2.6. Suppression of HBB Expression by Short Interfering RNA (siRNA)

siRNA was obtained from Integrated DNA Technologies (Coralville, IA, USA). The siRNA sequences were as follows: siHBB-1, 5′-GUG AAU UCU UUG CCA AAG UGA UGG GCC-3′; siHBB-2, 5′-AGU UUA GUA GUU GGA CUU AGG GAA CAA-3′. For the mock treatment, a non-specific control siRNA (siCont; Negative Control DsiRNA, Integrated DNA Technologies) was used. Transfection of siRNA into cells was performed using Lipofectamine RNAiMAX (Invitrogen) according to the manufacturer’s instructions, as previously described [23].

### 2.7. Cell Proliferation Assay

A498 cells were seeded at 2 × 10^5^/well in 96-well plates and incubated for 16 h at 37 °C. After incubation, cells were transfected with siRNA. Next, cells were incubated for 24, 48, 72, or 96 h at 37 °C and stained with CellTiter 96 Aqueous One Solution Cell Proliferation Assay (Promega, Madison, WI, USA) for 1 h according to the manufacturer’s instructions. The optical density of each well was measured at 490 nm using a microplate reader (iMark; Bio-Rad Laboratories, Hercules, California, USA).

### 2.8. Cell Invasion Assay

Forty-eight hours after siRNA transfection, cells (1 × 10^5^ cells/chamber) were suspended in serum-free RPMI1640 and transferred to the upper chamber of a Corning BioCoat Matrigel Invasion Chamber (# 354480; Corning International, Corning, NY, USA). The lower chamber was filled with RPMI1640 containing 10% FBS. After 24 h in a CO_2_ incubator, the cells that passed through the membrane to the lower chamber were fixed and stained with a crystal violet solution (0.05% crystal violet, 30% neutral formalin, 10% ethanol). The cells were counted by microscopic observation at 100× magnification. The average number of counted cells was calculated from three fields per chamber. The invasion ratio (% invasion) was calculated using the following formula: % invasion = number of cells that passed through the Matrigel chamber/number of cells that passed through the control insert membrane. The invasion index was determined using the following formula: Invasion index = % invasion of HBB expression-suppressed cells treated with siHBB-1 or siHBB-2/% invasion of mock-treated cells transfected with siCont.

### 2.9. ROS Measurement

Forty-eight hours after siRNA transfection, cells were incubated with 5 µM CellRox Green (ThermoFisher Scientific) at 37 °C for 30 min. Next, cells were washed thrice with PBS and collected using a cell detachment agent. After fixation with PBS containing 2.5% formalin, the fluorescence intensity of the cells was measured using a flow cytometer (FACSCantoII) and analyzed using FlowJo software version 6.5.7. The rate of increase in intracellular ROS was calculated using the following formula: Increase (%) of ROS = 100 × [(fluorescence intensity of HBB suppressed cells treated with siHBB-1 or siHBB-2) − (fluorescence intensity of mock-treated cells transfected with siCont)]/(fluorescence intensity of mock-treated cells transfected with siCont).

### 2.10. Statistical Analysis

All statistical analyses were performed using EZR version 1.35 (Saitama Medical Center, Jichi Medical University, Saitama, Japan) or GraphPad Prism version 5.03 (GraphPad Software, San Diego, CA, USA). EZR is a graphical interface for R software (R Foundation for Statistical Computing, Vienna, Austria) and is a modified version of R Commander version 2.3 [24]. The statistical processing methods are described in the figure legends. *p*-values ≤ 0.05 were considered statistically significant. The statistical methods used for the analyses are described in the tables and figure legends.

## 3. Results

### 3.1. HBB Expression in ccRCC Is Associated with Prognosis

To examine the clinical significance of HBB protein expression, immunohistochemical staining was performed on samples from 203 patients with non-metastatic ccRCC. Four cases of typical HBB-negative staining and four cases of typical HBB-positive staining are shown in Figure 1a,b, respectively. HBB expression was mainly detected in the cytoplasm of ccRCC cells but was not abundant in vessels or interstitial tissues.

Table 1 shows the relationship between HBB expression and clinicopathological patient characteristics. The median postoperative follow-up period was 8.12 years. A total of 121 radical and 82 partial nephrectomies were performed. Open surgery was performed in 110 patients and laparoscopic surgery was performed in 93 patients. HBB positivity or negativity was significantly associated with the T stage, N stage, grade, microvascular invasion, and infiltration type. HBB-positive patients had a significantly higher recurrence rate (*p* < 0.001) and significantly shorter cancer-specific survival (*p* = 0.00476) and overall survival (*p* = 0.015) than HBB-negative patients. 

In the Kaplan–Meier survival analysis, HBB-positive cases had significantly shorter recurrence-free (*p* = 0.000288, Figure 2a) and cancer-specific survival periods (*p* = 0.00381, Figure 2b) than those for HBB-negative cases. These results indicated that HBB expression in ccRCC is associated with prognosis. 

### 3.2. Several ccRCC Cell Lines Express HBB

HBB protein expression in ccRCC cell lines A498, Caki1, 769-P, and 786-O was examined using flow cytometry. HBB expression was not detected on the cell surface of any cell line (Figure 3a); however, HBB expression was detected in each cell line using the intracellular staining method (Figure 3b). A summary of the measurements with intracellular staining suggests that the four ccRCC cell lines expressed HBB (Figure 3c). Furthermore, mRNA expression of HBB was detected in the four cell lines using RT-PCR (Figure 4). These results indicated that these cell lines expressed HBB under normal culture conditions.

### 3.3. Suppression of HBB in ccRCC Cells Inhibits Both Cell Proliferation and Invasion

To verify the function of HBB in ccRCC cells, we suppressed HBB expression in A498 cells using HBB-specific siRNAs. As shown in Figure 5a, when we used siRNAs (siHBB-1 or siHBB-2), the expression of HBB mRNA was significantly suppressed after 24 h compared with that of the control siRNA (siCont). 

Cell proliferation ability was measured when HBB expression was suppressed using the conditions above. The cell proliferation ability of cells treated with siHBB-1 or siHBB-2 decreased significantly compared to that of cells treated with siCont (Figure 5b). These results suggest that HBB expression contributes to the enhancement of cell proliferation. 

We also confirmed the effect of HBB expression on cell invasion. In the absence of Matrigel, many cells passed through the membrane even after transfection with siHBB-1 or siHBB-2. In contrast, in the presence of Matrigel, the number of cells that passed through the membrane was significantly reduced after treatment with siHBB-1 or siHBB-2 (Figure 5c). The summary of cell invasion assay data showed that the cell invasion ability of the cells transfected with siHBB-1 or siHBB-2 was significantly decreased compared to that of the cells treated with siCont (Figure 5d). Taken together, these results indicated that HBB expression in ccRCC cells augmented both cell proliferation and invasion.

### 3.4. HBB Expression in ccRCC Cells Is Involved in Oxidative Adaptation

To clarify whether HBB expression in ccRCC cells is involved in the adaptation to oxidative stress, we investigated the relationship between ROS and HBB expression. First, we examined whether HBB expression levels were altered by oxidative stress. H_2_O_2_ was added to the culture medium of A498 cells to induce oxidative stress, and N-acetylcysteine (NAC) was used as an ROS scavenger. HBB expression levels were measured using flow cytometry 24 h after the addition of H_2_O_2_ and/or NAC. HBB expression levels were significantly augmented by the addition of H_2_O_2_, and the HBB expression levels were not changed by the combination of H_2_O_2_ and NAC (Figure 6a). These results suggest that oxidative stress augmented HBB expression levels in ccRCC cells.

Next, we measured whether HBB expression levels altered the ROS levels. The level of ROS increased in cells treated with siHBB-1 or siHBB-2 compared to that in cells treated with siCont (Figure 6b). These results suggest that HBB expression levels in ccRCC cells reduce ROS levels.

## 4. Discussion

This study revealed that HBB-positive ccRCC cases have a poor prognosis compared to that of HBB-negative cases. This is the first study to report the relationship between HBB and cancer prognosis using immunohistochemistry.

Hemoglobin is a protein present in erythrocytes, and its constituent, HBB, was thought to be expressed only in erythroid cells. However, in recent years, HBB expression has been detected in non-erythroid tissues such as neuronal cells [13], retinal epithelial cells [9], endometrial cells [25], cervical epithelial cells [4], type II alveolar epithelial cells [6,8], mesangial cells [10], hepatocytes [11], and tumor tissues [4,15,16,26,27]. These reports suggest that HBB functions as an ROS scavenger. Moreover, it has been reported that hemoglobin can remove ROS more efficiently than glutathione peroxidase, which is a known antioxidant [28].

ROS are produced as byproducts of aerobic metabolism in the mitochondria. Low levels of ROS act as signaling agents in cell proliferation and survival, moderate levels of ROS induce DNA mutations [29], and high levels of ROS cause cell death [30]. All cells regulate oxidative stress by producing antioxidants such as glutathione peroxidase. Tumor cells are exposed to higher levels of ROS than normal cells because of their highly proliferative and anchorage-independent growth [31]. The therapeutic effects of radiotherapy and certain anticancer drugs are mediated by the generation of ROS. These studies suggest that oxidative stress resistance mechanisms are essential for tumor cell survival, proliferation, and metastasis [16,32,33,34].

The relationship between tumors and HBB has been reported in several studies. In thyroid cancer [27] and neuroblastoma [26], HBB has been reported to have an inhibitory effect on tumors. In contrast, HBB has been reported to increase tumor activity in breast cancer [15,16], lung cancer [16], prostate cancer [16], and cervical cancer [4]. Thus, the effect of HBB on malignant tumor transformation has not been elucidated.

The molecular mechanism by which HBB is involved in tumor progression remains unknown. One potential molecular mechanism is the stabilization of transcription factor BACH1 via redox balance. Previously, the upregulation of BACH1 has been found to be associated with worsening prognoses in both lung and renal cancers [35,36,37]. In lung cancer, the upregulation of BACH1 is mediated by the inhibition of free heme via heme oxygenase-1 (HO-1) [35,37]. In contrast, HBB binds stably to heme [38], and therefore may also suppress free heme and increase BACH1 expression. However, in ccRCC, BACH1 expression regulates HO-1 expression, as in a feedback mechanism [36]. Further research is needed to determine how HBB is involved in the HO-1-mediated regulation of free heme and BACH1 in ccRCC.

ROS can both promote and suppress metastasis, suggesting that the mechanism by which ROS affect cancers is dependent on multiple observational conditions [39]. Thus, to determine whether HBB is a factor in cancer exacerbation, various molecular groups related to tumor exacerbation (Bcl-2 family and cadherins) should be examined as described previously [40]. Additionally, it is necessary to reconstruct clinical results using animal experiments to reach comprehensive conclusions on HBB function. However, because HBB reduction suppresses cell proliferation, it is assumed that a genetic control mechanism that can turn HBB expression on or off in vivo after transplantation of cell lines will be required to establish animal experiments. In the future, we aim to elucidate the function of HBB using the on or off toggle system of HBB expression.

Some points still need to be clarified to control HBB expression and apply it to tumor therapy. First, the mechanism by which HBB enhances cell proliferation and invasion remains unclear. It is thought that the reduction of ROS by HBB prevents the functional deterioration of proteins involved in cell proliferation and invasion; however, the changes in the signaling pathways involved in proliferation and invasion remain unclear. In particular, the relationship between HIF1α, which is induced by hypoxic responses and has a key role in tumor malignancy, and HBB is not well understood. Second, it is unclear whether HBB exists as a monomer or multimer with heme and other globin chains. To the best of our knowledge, there are no reports describing the multimerization of HBB in tumor tissues. Third, the transcriptional mechanisms of HBB in tumor cells remain unclear. For example, in alveolar epithelial cells, such as erythroid cells, transcription of the *HBB* gene is regulated by GATA-1 [7], but this is not the case in cervical cancer [4]. Kruppel-like factor 4 (KLF4) and NF-E2-related factor 2 (NRF2) are also involved in the transcriptional regulation of *HBB* [16,41]. KLF4 regulates *HBB* transcription in circulating tumor cells, and constitutively active mutations in *NRF2* have been reported in various carcinomas, including RCC [42]. Further studies are needed to determine the mechanisms by which KLF4 and NRF2 regulate *HBB* transcription in RCC cells. Finally, for prognostic purposes, the relationship between mRCC and HBB expression should be investigated.

Herein, we measured the decrease in the HBB protein level by flow cytometry after siHBB treatment. However, a stable decrease in HBB expression was not observed in the harvested cells (data not shown) [43]. One reason as to why HBB down-expression was not observed may be related to the fact that HBB down-expression reduces cell proliferation, thus reducing the proportion of down-expressed HBB cells. Alternatively, HBB stability might last longer when HBB production is reduced for cell survival. Because the binding between heme and HBB is stable [38], cancer cells that constitutively produce free heme may require constant HBB synthesis for heme binding. In the future, it will be necessary to investigate the turnover of free heme and HBB.

In this study, patients with HBB-positive ccRCC were found to have a poorer prognosis compared to HBB-negative patients. HBB is believed to enhance ccRCC activity by functioning as an ROS scavenger. As with other carcinomas, investigating the expression of HBB in ccRCC would help in the discovery of new prognostic factors.

## Figures and Tables

**Figure 1 biomedicines-11-01330-f001:**
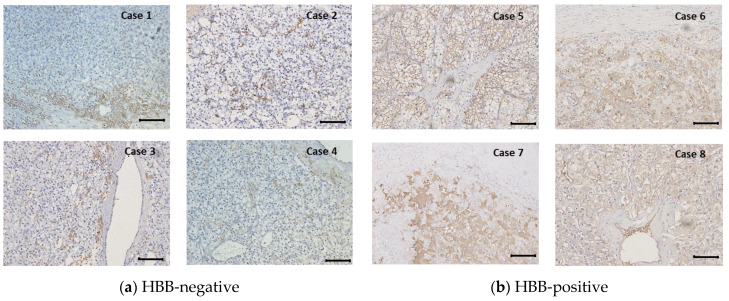
Immunohistochemical observation of hemoglobin β (HBB) expression in non-metastatic clear cell renal cell carcinoma (ccRCC) tissue. Typical HBB-negative ((**a**); Cases 1–4) and HBB-positive ((**b**); Cases 5–8) tissues are shown. Bar, 100 μm.

**Figure 2 biomedicines-11-01330-f002:**
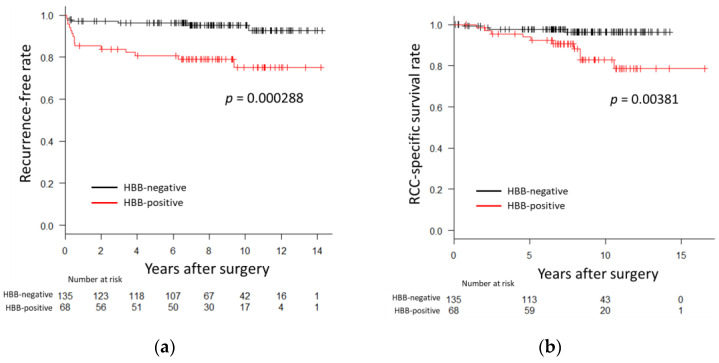
Prognosis and hemoglobin β (HBB) expression in non-metastatic clear cell renal cell carcinoma (ccRCC). Kaplan–Meier analyses were performed on the recurrence-free rate (**a**) and cancer-specific survival rate (**b**) in 203 cases of non-metastatic ccRCC. Black lines indicate HBB-negative cases and red lines indicate HBB-positive cases. *p* values were calculated using the long-rank test.

**Figure 3 biomedicines-11-01330-f003:**
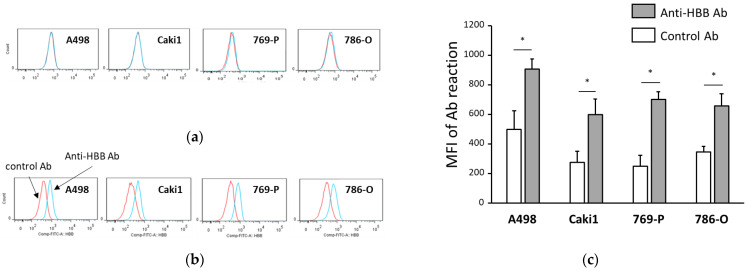
Measurement of hemoglobin β (HBB) expression in clear cell renal cell carcinoma (ccRCC) cell lines using flow cytometry. Four ccRCC cell lines, A498, Caki1, 769-P, and 786-O, were used for the flow cytometric measurements. The cells were stained with FITC-conjugated antibodies before (**a**) or after (**b**) cell membrane permeabilization. Red line histograms indicate the results of staining with isotype-matched control antibodies, and blue line histograms show the results of staining with anti-HBB antibodies. A summary of the results is shown in (**c**). The vertical axis indicates the mean fluorescence intensity (MFI) of each antibody reaction, and the horizontal axis indicates the measured cell lines. Open columns indicate the reaction with the control antibody, and closed columns are the result of the anti-HBB antibody reaction. Data represent the mean ± standard error from three or more independent measurements. *p* values were determined by a paired student *t*-test (two-tailed). * *p* < 0.05.

**Figure 4 biomedicines-11-01330-f004:**
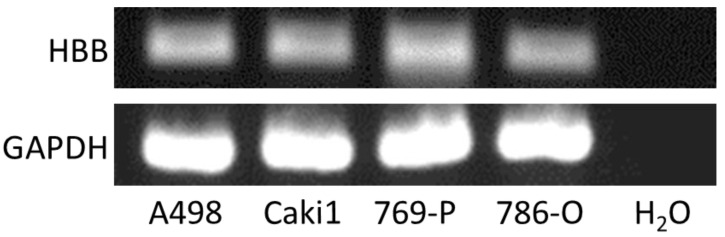
Detection of hemoglobin β (HBB) mRNA expression in clear cell renal cell carcinoma (ccRCC) cell lines by RT-PCR. Total RNAs were purified from four ccRCC cell lines (A498, Caki1, 769-P, and 786-O) and used in a reverse transcription reaction to synthesize cDNA. H_2_O was used as a control to show the background amplification. The HBB PCR products (250 bp) and the glyceraldehyde-3-phosphate dehydrogenase (GAPDH) PCR products (138 bp) are shown. A representative result from two independent experiments is shown.

**Figure 5 biomedicines-11-01330-f005:**
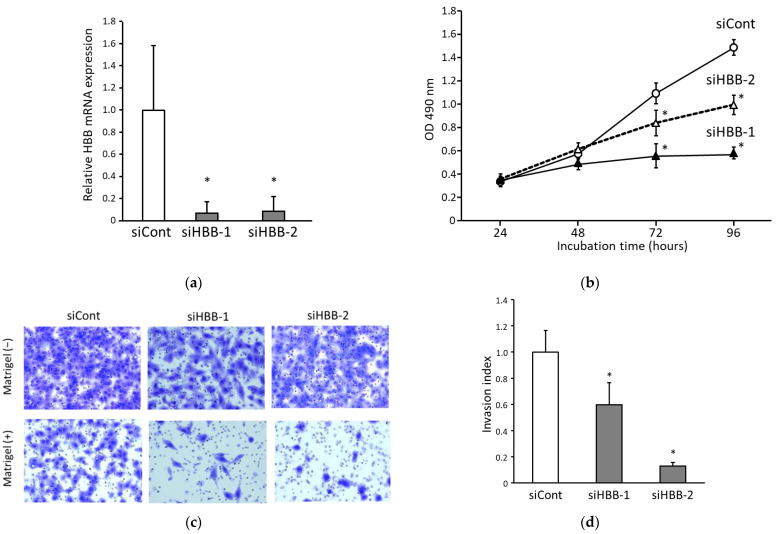
Contribution of hemoglobin β (HBB) expression to the malignant transformation of clear cell renal cell carcinoma (ccRCC) cells. (**a**) Suppression of HBB mRNA expression by siRNA transfection. siHBB-1 or siHBB-2 was transfected into A498 cells, and control siRNA (siCont) was transfected as a mock treatment. The HBB mRNA expression levels in the treated cells with transfected siRNA were measured by quantitative real-time RT-PCR. Relative HBB mRNA expression was calculated based on glyceraldehyde-3-phosphate dehydrogenase (GAPDH) mRNA expression. Data represent the mean ± standard deviation from five independent experiments. *p* values were analyzed using one-way analysis of variance (ANOVA) with a post hoc test using Dunnett and compared to the cells transfected with siCont. (**b**) Cell proliferation assay of A498 cells transfected with siRNA. After the siRNA transfection, the cells were incubated for 24–96 h. Data represent the mean ± standard deviation from three independent experiments. *p* values were analyzed using one-way ANOVA with a post hoc test using Holm and compared to the siCont-transfected cells at each incubation time. (**c**) Representative photographs of transmigrated A498 cells passed through the membrane after the cell invasion assay. Upper panels are control chamber membranes, shown as Matrigel (−). Bottom panels indicate the Matrigel chamber, shown as Matrigel (+). Left-side panels are the cells transfected with siCont, middle panels are transfected with siHBB-1, and right-side panels are transfected with siHBB-2 (magnification, ×200). A summary of the invasion indexes is shown in (**d**). Data are the mean ± standard deviation from three independent experiments. *p* values were analyzed using one-way ANOVA with a post hoc test using Dunnett and compared to the siCont-transfected cells. * *p* < 0.05.

**Figure 6 biomedicines-11-01330-f006:**
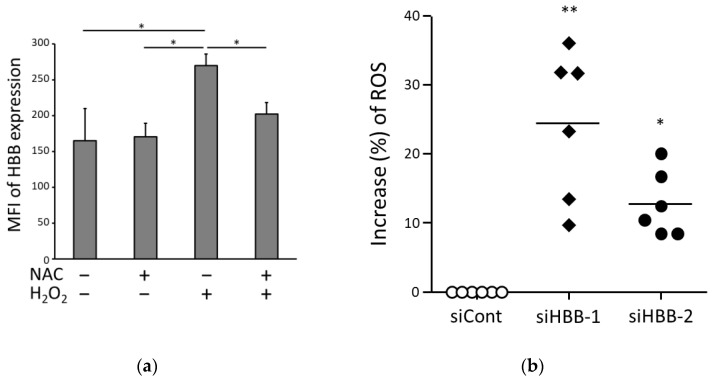
Interaction between hemoglobin β (HBB) expression and oxidative stress. (**a**) HBB expression due to oxidative stress. A498 cells were incubated with (+) or without (−) H_2_O_2_ (2 mM) and/or N-acetyl cysteine (NAC; 10 mM) for 24 h. The level of HBB expression was measured via flow cytometry, as described in Figure 3b. Data show the mean ± standard deviation from four independent experiments. *p* values were analyzed via one-way analysis of variance (ANOVA) with a post hoc test using Bonferroni, * *p* < 0.05. (**b**) A498 cells were transfected with siRNA (siCont, open circle; siHBB-1, closed diamond; siHBB-2, closed circle) and incubated for 48 h. After the incubation, the amount of ROS in the cells was measured with CellRox green staining. Dot plots and the means from six independent experiments are shown as bars. *p* values were calculated using non-parametric one-way ANOVA (Friedman tests) with a post hoc test with Dunn, compared to the treatment with siCont. ** *p* < 0.01, * *p* < 0.05.

**Table 1 biomedicines-11-01330-t001:** Clinicopathological backgrounds of study patients, and associations with HBB expression.

Factor	All Patients		HBB		
			Negative	Positive	*p* Value
**Number of patients**	203		135	68	
**Age at surgery**	Mean ± SD	63.51 ± 11.58	63.2 ± 11.7	64.1 ± 11.4	0.61 ^b^
	Range	28–85	28–82	36–85	
**Sex**	Male	125	86 (68.8)	39 (31.2)	0.468 ^a^
	Female	78	49 (62.8)	29 (37.2)	
**Tumor laterality**	Left	95	64 (67.4)	31 (32.6)	0.923 ^a^
	Right	108	71 (65.7)	37 (34.3)	
**Pathological T stage**	pT1a	124	100 (74.1)	24 (35.3)	<0.001 ^a^
	pT1b	40	19 (14.1)	21 (30.9)	
	pT2a	8	2 (1.5)	6 (8.8)	
	pT2b	6	4 (3.0)	2 (2.9)	
	pT3a	15	7 (5.2)	8 (11.8)	
	pT3b	7	2 (1.5)	5 (7.4)	
	pT3c	1	0 (0.0)	1 (1.5)	
	pT4	2	1 (0.7)	1 (1.5)	
**Pathological N stage**	pN0/pNX	199 (98.0)	134 (99.3)	65 (86.8)	0.0021 ^a^
	pN1	4 (2.0)	1 (0.7)	3 (4.4)	
**Grade**	G1	96	80 (59.3)	16 (23.5)	<0.001 ^a^
	G2	82	48 (35.6)	34 (50.0)	
	G3	20	4 (3.0)	16 (23.5)	
	G4	5	3 (2.2)	2 (2.9)	
**Microvascular invasion**	Negative	180 (88.6)	127 (70.6)	53 (29.4)	0.00143 ^a^
	Positive	23 (11.4)	8 (34.8)	15 (65.2)	
**INF**	INFa	171 (84.2)	121 (89.6)	50 (73.5)	0.00566 ^a^
	INFb	32 (15.8)	14 (10.4)	18 (26.5)	
**Event**					
**Recurrence**	No	181	128 (70.7)	53 (29.3)	<0.001 ^a^
	Yes	22	7 (31.8)	15 (68.2)	
**Cancer specific survival**	Alive	189	131 (69.3)	58 (30.7)	0.00476 ^a^
	Deceased	14	4 (28.6)	10 (71.4)	
**Overall survival**	Alive	169	119 (70.4)	50 (29.6)	0.015 ^a^
	Deceased	34	16 (47.1)	18 (52.9)	
**Follow-up Years**	Median	8.12	8.06	8.24	0.797 ^c^
	(IQR)	(6.56–10.72)	(6.52–10.79)	(6.69–10.69)	

Abbreviations: HBB, hemoglobin; β INF, infiltration type; SD, standard distribution; IQR, interquartile range, Note: ^a^, Chi-square test; ^b^, Student’s *t*-test; ^c^, Mann–Whitney U test. Number in parentheses indicate % unless otherwise specified.

## Data Availability

The data supporting the findings of this study are available from the corresponding author, Y.T., upon reasonable request.

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
