# Peer review of "Hemoglobin β Expression Is Associated with Poor Prognosis in Clear Cell Renal Cell Carcinoma"

_biomedicines, 2023, doi:10.3390/biomedicines11051330_

Round 1

Reviewer 1 Report

In this manuscript, the authors have made an attempt to investigate the role of 

 hemoglobin β-chain (HBB) in renal cancer. They have concluded from the IHC data that HBB-positive ccRCC cases have a poor prognosis compared of HBB-negative cases. Although this has been extensively studied in other cancer types, this manuscript reports for the first time that increased HBB  has a poor prognosis in renal cancer.

In vitro studies suggests that knockdown of HBB in renal cancer cell lines show decreased cell proliferation and migration. 

Although, the initial data is convincing, the manuscript considerably lacks molecular mechanism and animal experiments. The studies are in a very preliminary stage and needs more studies to support the hypothesis.

Author Response

Comment of Reviewer-1

In this manuscript, the authors have made an attempt to investigate the role of hemoglobin β-chain (HBB) in renal cancer. They have concluded from the IHC data that HBB-positive ccRCC cases have a poor prognosis compared of HBB-negative cases. Although this has been extensively studied in other cancer types, this manuscript reports for the first time that increased HBB has a poor prognosis in renal cancer.

In vitro studies suggests that knockdown of HBB in renal cancer cell lines show decreased cell proliferation and migration. 

Although, the initial data is convincing, the manuscript considerably lacks molecular mechanism and animal experiments. The studies are in a very preliminary stage and needs more studies to support the hypothesis.

Response to reviewer 1

Thank you for the time and effort that you have dedicated to providing your valuable feedback on our manuscript. We are grateful to you for your insightful comments on our paper. 

As the reviewer pointed out, experimental results on molecular mechanisms and comprehensive results from animal studies are needed to conclude the effects of HBB on clear renal cell carcinoma. We therefore changed our conclusions. By changing the conclusion, we have clarified that this is an early-stage research. Thus, we have revised the manuscript as follows:

The title and the conclusion of each section (Abstract, Introduction, and Discussion) have been revised as follows:

Title:  Hemoglobin β Expression is associated with Poor Prognosis in Clear Cell Renal Cell Carcinoma

Abstract:  HBB expression may serve as a new prognostic biomarker for ccRCC in the future.

Introduction: Furthermore, we examined the function of HBB in RCC cell lines to examine the possibility of HBB as a prognostic biomarker.

Discussion: As with other carcinomas, investigating the expression of HBB in ccRCC should be useful to discover new prognostic factors.

Furthermore, in accordance with the reviewer's indications, we added the following discussion to the assumed molecular mechanism to supplement the lack of findings, as follows:

1)      The molecular mechanism by which HBB is involved in tumor progression remains unknown. One potential molecular mechanism is the stabilization of transcription factor BACH1 via redox balance. Previously, the upregulation of BACH1 has been found to be associated with worsening prognoses in both lung and renal cancers [35-37]. In lung cancer, the upregulation of BACH1 is mediated by the inhibition of free heme via heme oxygenase-1 (HO-1) [35,36]. In contrast, HBB binds stably to heme [38], and therefore may also suppress free heme and increase BACH1 expression. However, in ccRCC, BACH1 expression regulates HO-1 expression, as in a feedback mechanism [37]. Further research is needed to determine how HBB is involved in the HO-1-mediated regulation of free heme and BACH1 in ccRCC.

2)       ROS can both promote and suppress metastasis, suggesting that the mechanism by which ROS affects cancers is dependent on multiple observational conditions [39]. Thus, to determine whether HBB is a factor in cancer exacerbation, various molecular groups related to tumor exacerbation (Bcl-2 family and cadherins) should be examined as described previously [40]. Additionally, it is necessary to reconstruct clinical results using animal experiments to reach comprehensive conclusions on HBB function. However, because HBB reduction suppresses cell proliferation, it is assumed that a genetic control mechanism that can turn HBB expression on or off in vivo after transplantation of cell lines will be required to establish animal experiments. In the future, we aim to elucidate the function of HBB using the on or off toggle system of HBB expression.

Added references:

35.          Lignitto, L.; LeBoeuf, S. E.; Homer, H.; Jiang, S.; Askenazi, M.; Karakousi, T. R.; Pass, H. I.; Bhutkar, A. J.; Tsirigos, A.; Ueberheide, B.; Sayin, V. I.; Papagiannakopoulos, T.; Pagano, M., Nrf2 Activation Promotes Lung Cancer Metastasis by Inhibiting the Degradation of Bach1. Cell 2019, 178, (2), 316-329 e18.

36.          Wiel, C.; Le Gal, K.; Ibrahim, M. X.; Jahangir, C. A.; Kashif, M.; Yao, H.; Ziegler, D. V.; Xu, X.; Ghosh, T.; Mondal, T.; Kanduri, C.; Lindahl, P.; Sayin, V. I.; Bergo, M. O., BACH1 Stabilization by Antioxidants Stimulates Lung Cancer Metastasis. Cell 2019, 178, (2), 330-345 e22.

37.          Takemoto, K.; Kobatake, K.; Miura, K.; Fukushima, T.; Babasaki, T.; Miyamoto, S.; Sekino, Y.; Kitano, H.; Goto, K.; Ikeda, K.; Hieda, K.; Hayashi, T.; Hinata, N.; Kaminuma, O., BACH1 promotes clear cell renal cell carcinoma progression by upregulating oxidative stress-related tumorigenicity. Cancer Sci 2023, 114, (2), 436-448.

38.          Gattoni, M.; Boffi, A.; Sarti, P.; Chiancone, E., Stability of the heme-globin linkage in alphabeta dimers and isolated chains of human hemoglobin. A study of the heme transfer reaction from the immobilized proteins to albumin. J Biol Chem 1996, 271, (17), 10130-6.

39.          Cheung, E. C.; Vousden, K. H., The role of ROS in tumour development and progression. Nat Rev Cancer 2022, 22, (5), 280-297.

40.          Ding, Y.; Xiong, S.; Chen, X.; Pan, Q.; Fan, J.; Guo, J., HAPLN3 inhibits apoptosis and promotes EMT of clear cell renal cell carcinoma via ERK and Bcl-2 signal pathways. J Cancer Res Clin Oncol 2023, 149, (1), 79-90.

Reviewer 2 Report

Kurota et.al have hypothesized a pro-tumorigenic role for Hemoglobin-b HBB in renal cancer, based on other reports on increased expression of HBB in other cancer types. HBB binds oxyge and the very expression of HBB in cell types other than erythrocytes, where it is primarly expressed, suggests that HBB can function to bind reactive oxygen species (which can kill cancer cells).

The authors have used clinical samples of renal cell carcinoma (RCC) patients and analyze the expression of HBB and correlated with survival. IN VITRO, the authors have demonstarted that HBB plays a role in renar cancer cell proliferation and inhibiting ROS. Together, HBB serves as a novel therapeutic target for RCC.

However some points arise:

1 . Proliferation and invasion markers (like WB of proteins involved would be better. Reduced anti-apoptotic markers bcI-2, reduced HO-I which plays a major role in RCC and redox, levels of metastatic markers like N/C-Cadherin) are needed to support the finding.

2.only mRNA expression for siRNA inhibition is shown. At the protein level Flow or WB is needed)

Although the patient data looks just fine, the manuscript needs some additional data in support of their in vitro conclusions as discussed in the points above.

Author Response

Comment of Reviewer-2

Kurota et.al have hypothesized a pro-tumorigenic role for Hemoglobin-b HBB in renal cancer, based on other reports on increased expression of HBB in other cancer types. HBB binds oxyge and the very expression of HBB in cell types other than erythrocytes, where it is primarly expressed, suggests that HBB can function to bind reactive oxygen species (which can kill cancer cells).

The authors have used clinical samples of renal cell carcinoma (RCC) patients and analyze the expression of HBB and correlated with survival. IN VITRO, the authors have demonstarted that HBB plays a role in renar cancer cell proliferation and inhibiting ROS. Together, HBB serves as a novel therapeutic target for RCC.

However some points arise:

1 . Proliferation and invasion markers (like WB of proteins involved would be better. Reduced anti-apoptotic markers bcI-2, reduced HO-I which plays a major role in RCC and redox, levels of metastatic markers like N/C-Cadherin) are needed to support the finding.

2.only mRNA expression for siRNA inhibition is shown. At the protein level Flow or WB is needed)

Although the patient data looks just fine, the manuscript needs some additional data in support of their in vitro conclusions as discussed in the points above.

Response to reviewer 2

Thank you for the time and effort that you have dedicated to providing your valuable feedback on our manuscript. We are grateful to you for your insightful comments on our paper.

As the reviewer pointed out, additional experimental results on molecular mechanisms are needed to conclude the effects of HBB on clear renal cell carcinoma. We therefore changed our conclusions. By changing the conclusion, we have clarified that this is an early-stage research. Thus, we have revised the manuscript as follows:

The title and the conclusion of each section (Abstract, Introduction, and Discussion) have been revised as follows:

Title:  Hemoglobin β Expression is associated with Poor Prognosis in Clear Cell Renal Cell Carcinoma

Abstract:  HBB expression may serve as a new prognostic biomarker for ccRCC in the future.

Introduction: Furthermore, we examined the function of HBB in RCC cell lines to examine the potential of HBB as a prognostic biomarker.

Discussion: As with other carcinomas, investigating the expression of HBB in ccRCC would help in the discovery of new prognostic factors.

Furthermore, we revised the discussion as follows:

Answer to comment 1

As the reviewer pointed out, it is necessary to show experimental and comprehensive results on the molecular mechanism to identify the action of HBB on ccRCC. Therefore, the following text was added to the Discussion section.

The molecular mechanism by which HBB is involved in tumor progression remains unknown. One potential molecular mechanism is the stabilization of transcription factor BACH1 via redox balance. Previously, the upregulation of BACH1 has been found to be associated with worsening prognoses in both lung and renal cancers [35-37]. In lung cancer, the upregulation of BACH1 is mediated by the inhibition of free heme via heme oxygenase-1 (HO-1) [35, 36]. In contrast, HBB binds stably to heme [38], and therefore may also suppress free heme and increase BACH1 expression. However, in ccRCC, BACH1 expression regulates HO-1 expression, as in a feedback mechanism [37]. Further research is needed to determine how HBB is involved in the HO-1-mediated regulation of free heme and BACH1 in ccRCC.

ROS can both promote and suppress metastasis, suggesting that the mechanism by which ROS affects cancers is dependent on multiple observational conditions [39]. Thus, to determine whether HBB is a factor in cancer exacerbation, various molecular groups related to tumor exacerbation (Bcl-2 family and cadherins) should be examined as described previously [40]. Additionally, it is necessary to reconstruct clinical results using animal experiments to reach comprehensive conclusions on HBB function. However, because HBB reduction suppresses cell proliferation, it is assumed that a genetic control mechanism that can turn HBB expression on or off in vivo after transplantation of cell lines will be required to establish animal experiments. In the future, we aim to elucidate the function of HBB using the on or off toggle system of HBB expression.

Answer to comment 2

As the Reviewer pointed out, we thought it necessary to confirm the reduction at the protein level by siHBB. However, in this study, it was difficult to constitutively control HBB protein expression levels. We have added the following text to the manuscript:

 Herein, we measured the decrease in the HBB protein level using flow cytometry after siHBB treatment. However, a stable decrease in HBB expression was not observed in the harvested cells (data not shown; please see below). One reason as to why HBB down-expression was not observed may be related to the fact that HBB down-expression reduces cell proliferation, thus reducing the proportion of down-expressed HBB cells. Alternatively, HBB stability might last longer when HBB production is reduced for cell survival. Because the binding between heme and HBB is stable [38], cancer cells that constitutively produce free heme may require constant HBB synthesis for heme binding. In the future, it will be necessary to investigate the turnover of free heme and HBB.

A498 cells were treated with siRNA (siCont, siHBB-1, or siHBB-2) as described in the Materials and Methods section. After 72 h, the expression of HBB in the cells were analyzed using flow cytometry similar to that described in Figure 3. This figure is not included in the manuscript because we thought it would confuse the reader's understanding. If you recommend adding this data as Supplementary Data, please let us know and I will add it to the Supplementary Data.

Added references:

  1. Lignitto, L.; LeBoeuf, S. E.; Homer, H.; Jiang, S.; Askenazi, M.; Karakousi, T. R.; Pass, H. I.; Bhutkar, A. J.; Tsirigos, A.; Ueberheide, B.; Sayin, V. I.; Papagiannakopoulos, T.; Pagano, M., Nrf2 Activation Promotes Lung Cancer Metastasis by Inhibiting the Degradation of Bach1. Cell 2019, 178, (2), 316-329 e18.
  2. Wiel, C.; Le Gal, K.; Ibrahim, M. X.; Jahangir, C. A.; Kashif, M.; Yao, H.; Ziegler, D. V.; Xu, X.; Ghosh, T.; Mondal, T.; Kanduri, C.; Lindahl, P.; Sayin, V. I.; Bergo, M. O., BACH1 Stabilization by Antioxidants Stimulates Lung Cancer Metastasis. Cell 2019, 178, (2), 330-345 e22.
  3. Takemoto, K.; Kobatake, K.; Miura, K.; Fukushima, T.; Babasaki, T.; Miyamoto, S.; Sekino, Y.; Kitano, H.; Goto, K.; Ikeda, K.; Hieda, K.; Hayashi, T.; Hinata, N.; Kaminuma, O., BACH1 promotes clear cell renal cell carcinoma progression by upregulating oxidative stress-related tumorigenicity. Cancer Sci 2023, 114, (2), 436-448.
  4. Gattoni, M.; Boffi, A.; Sarti, P.; Chiancone, E., Stability of the heme-globin linkage in alphabeta dimers and isolated chains of human hemoglobin. A study of the heme transfer reaction from the immobilized proteins to albumin. J Biol Chem 1996, 271, (17), 10130-6.
  5. Cheung, E. C.; Vousden, K. H., The role of ROS in tumour development and progression. Nat Rev Cancer 2022, 22, (5), 280-297.
  6. Ding, Y.; Xiong, S.; Chen, X.; Pan, Q.; Fan, J.; Guo, J., HAPLN3 inhibits apoptosis and promotes EMT of clear cell renal cell carcinoma via ERK and Bcl-2 signal pathways. J Cancer Res Clin Oncol 2023, 149, (1), 79-90.

Round 2

Reviewer 1 Report

The authors have addressed the comments. However, the studies are still in a very preliminary stage and needs some more studies in the future to strengthen it.